# Prognostic Role of Circulating miRNAs in Early-Stage Non-Small Cell Lung Cancer

**DOI:** 10.3390/jcm8020131

**Published:** 2019-01-23

**Authors:** Paola Ulivi, Elisabetta Petracci, Giorgia Marisi, Sara Baglivo, Rita Chiari, Monia Billi, Matteo Canale, Luigi Pasini, Serena Racanicchi, Alessandro Vagheggini, Angelo Delmonte, Marita Mariotti, Vienna Ludovini, Massimiliano Bonafè, Lucio Crinò, Francesco Grignani

**Affiliations:** 1Biosciences Laboratory, Istituto Scientifico Romagnolo per lo Studio e la Cura dei Tumori (IRST) IRCCS, 47014 Meldola, Italy; giorgia.marisi@irst.emr.it (G.M.); matteo.canale@irst.emr.it (M.C.); luigi.pasini@irst.emr.it (L.P.); massimiliano.bonafe@irst.emr.it (M.B.); 2Biostatistics and Clinical Trials Unit, Istituto Scientifico Romagnolo per lo Studio e la Cura dei Tumori (IRST) IRCCS, 47014 Meldola, Italy; elisabetta.petracci@irst.emr.it (E.P.); alessandro.vagheggin@irst.emr.it (A.V.); 3Department of Medical Oncology, Santa Maria della Misericordia Hospital, Azienda Ospedaliera di Perugia, 06129 Perugia, Italy; baglivosara@gmail.com (S.B.); rita.chiari@ospedale.perugia.it (R.C.); vienna.ludovini@ospedale.perugia.it (V.L.); 4General Pathology Unit, Department of Medicine, University of Perugia, 06132 Perugia, Italy; monia.billi@unipg.it (M.B.); serena.racanicchi@unipg.it (S.R.); francesco.grignani@unipg.it (F.G.); 5Department of Medical Oncology, Istituto Scientifico Romagnolo per lo Studio e la Cura dei Tumori (IRST) IRCCS, 47014 Meldola, Italy; angelo.delmonte@irst.emr.it (A.D.); marita.mariotti@irst.emr.it (M.M.); lucio.crino@irst.emr.it (L.C.)

**Keywords:** miRNAs, early-stage NSCLC, prognosis, plasma

## Abstract

Non-small cell lung cancer (NSCLC) is the primary cause of cancer-related death worldwide, with a low 5-year survival rate even in fully resected early-stage disease. Novel biomarkers to identify patients at higher risk of relapse are needed. We studied the prognostic value of 84 circulating microRNAs (miRNAs) in 182 patients with resected early-stage NSCLC (99 adenocarcinoma (ADC), 83 squamous cell carcinoma (SCC)) from whom peripheral blood samples were collected pre-surgery. miRNA expression was analyzed in relation to disease-free survival (DFS) and overall survival (OS). In univariable analyses, five miRNAs (miR-26a-5p, miR-126-3p, miR-130b-3p, miR-205-5p, and miR-21-5p) were significantly associated with DFS in SCC, and four (miR-130b-3p, miR-26a-5p, miR-126-3p, and miR-205-5p) remained significantly associated with OS. In ADC, miR-222-3p, miR-22-3p, and mir-93-5p were significantly associated with DFS, miR-22-3p remaining significant for OS. Given the high-dimensionality of the dataset, multivariable models were obtained using a regularized Cox regression including all miRNAs and clinical covariates. After adjustment for disease stage, only miR-126-3p showed an independent prognostic role, with higher values associated with longer DFS in SCC patients. With regard to ADC and OS, no miRNA remained significant in multivariable analysis. Further investigation into the role of miR-126 as a prognostic marker in early-stage NSCLC is warranted.

## 1. Introduction

Non-small cell lung cancer (NSCLC) is the primary cause of cancer-related death worldwide [1] and only around 20% of patients present with early stage disease that can be resected [2]. However, of those who undergo surgery, approximately 50% relapse, mainly with distant metastases [3]. The principal clinical determinant in the prognosis of NSCLC is tumor extension, characterized by stage. However, there is substantial variability in disease outcome in patients sharing the same clinical features [4], suggesting that the management of NSCLC could be improved by combining the evaluation of molecular biomarkers with traditional cancer staging.

MicroRNAs (miRNAs) are small noncoding RNAs that post-transcriptionally regulate the translation of target genes and influence a series of cellular functions such as proliferation, differentiation and apoptosis. They may act as oncogenes or tumor suppressors having regulatory functions on multiple downstream genes with different biological activities and, as a consequence, they may provide a more accurate prediction of survival than the expression of a single marker or a gene expression profile alone [5]. Their expression is altered in all human malignancies [6,7] and is associated with prognosis and therapeutic outcome [8]. In lung cancer, some studies have demonstrated a potential role of different tumor tissue miRNAs in relation to outcome and prognosis of patients with lung adenocarcinoma (ADC) or squamous cell lung carcinoma (SCC) [9,10,11,12,13]. In operable NSCLC, some tissue miRNAs have been significantly associated with distant relapse and/or overall survival in SCC and/or ADC patients, suggesting their potential role in lung cancer growth and progression [12,14,15,16]. A miRNA signature composed of different miRNAs was also identified as being able to distinguish between stage I ADC patients who relapsed within two years of surgery and those who were still disease-free after three years [17,18]. As miRNAs are highly stable due to their resistance to endogenous and exogenous RNase activity and to extreme temperatures, extremes of pH (pH 1 or 13), extended storage in frozen conditions, and repeated freeze-thaw cycles [19], they represent ideal markers to be evaluated in biological fluids such as serum and plasma. Within the context of surgically resected NSCLC, some studies have reported a role of circulating miRNAs as potential prognostic factors. The possibility to detect and analyze miRNAs in blood gives the opportunity to avoid the necessity of the tumor tissue for the molecular analysis and to eventually monitor the levels of miRNAs during the patient’s follow-up. Some evidences have reported that serum levels of specific miRNAs in patients with early-stage NSCLC, analyzed at baseline before the surgical resection, were significantly associated with OS and prognosis [20,21].

In the present study we focused on early-stage (IA-IIIA) NSCLC and used real-time PCR to evaluate a panel of serum miRNAs as potential prognostic biomarkers in patients undergoing surgery. To achieve this purpose we took advantage of statistical methods for high-dimensional potentially correlated data. Moreover, cross-validation was used to evaluate models performance. 

## 2. Materials and Methods

### 2.1. Case Series

One-hundred-and-eighty-two patients with surgically resected stage IA-IIIA NSCLC (83 with SCC and 99 with ADC) were enrolled by the Medical Oncology Department of Perugia Hospital and by the Thoracic Surgery Department of Morgagni-Pierantoni Hospital of Forlì between February 2002 and January 2012. The study was approved by the Ethical Committee of both centers taking part (Perugia Hospital and IRST IRCCS; protocol IRST B058 26/07/2016) and written informed consent was obtained from all patients. The study was also conducted in accordance with the Declaration of Helsinki.

### 2.2. miRNA Expression Analysis

A peripheral blood sample was collected from each patient in tubes without anticoagulant the day before surgery and processed to obtain serum. Serum was stored at −80°C until miRNA extraction, and no previous freeze/thaw cycles were performed before extraction. Extraction of miRNAs was performed using miRNeasy Serum/Plasma Kit (Qiagen) starting from 200 µL of serum. The synthetic *C. elegans* syn-cel-miR-39 was used as a spike-in control during each extraction procedure. Five microliters of extracted miRNAs was retrotranscribed using miScript II RT Kit (Qiagen). A panel of 84 miRNAs was analyzed in single by real-time PCR using the Human Serum & Plasma miScript miRNA PCR Array (MIHS-106Z) (Qiagen), in which 84 pathway-specific miRNAs, 6 housekeeping snRNAs, 2 miRNA reverse transcription controls, 2 positive PCR controls, and 2 miRNA isolation controls were singly spotted in the array.

qRT-PCR data were normalized using one external spike-in (cel-miR-39) and 2 endogenous reference genes selected by NormFinder software. The difference between the cycle threshold (Ct) value of cel-miR-39 and its mean in all patients was subtracted from each Ct value of the target miRNA. From the quantity thus obtained, the mean of the Ct values of the two reference genes was subtracted (*ΔCT*). All statistical analyses were performed using relative miRNA expression value reported as *−ΔCT*. Both the spike-in and the 2 reference genes were not used in the statistical analyses but only for data normalization.

### 2.3. Statistical Analysis

Data were summarized as mean ± standard deviation (SD) for continuous variables and as counts and percentages for categorical variables. The main end point was disease-free survival (DFS), defined as the time since surgery until the date of disease relapse or death from any cause, whichever occurred first. Patients not experiencing any event were censored at the date of the last follow-up update in May 2016. The secondary end point was overall survival (OS) defined as the time since surgery until the date of death from any cause. Time was censored at the last follow-up if no event was reported.

We excluded miRNAs with >50% undetermined values from the analysis. The remaining values were imputed using the K-Nearest Neighbor method. Before imputation, Pearson’s chi-square test or Fisher’s Exact test (as appropriate) was used to test for a different proportion of undetermined values between patients with or without the events of interest. Moreover, before excluding such miRNAs from further analyses, an additional sensitivity analysis was performed. That is, evaluating their prognostic role in a univariate Cox model with and without imputed values.

All statistical analyses were performed separately for ADC and SCC patients and for the 2 time-to-event end points, DFS and OS. The Kaplan–Meier (K–M) method and log-rank test were used to compare DFS or OS between patient groups defined by covariates’ levels. Median DFS and corresponding 95% confidence intervals (CIs) are reported. Median follow-up time was computed as the median of times at risk for censored patients. Univariate Cox regression models were used to evaluate the direction and magnitude of the association between covariates, normalized miRNAs, and the outcome measures, i.e., DFS and OS. Analyses were performed with and without imputed data to evaluate the potential effect of the imputation procedure for miRNA values. As results (data not shown) were very similar, we decided to report only those from imputed data. The Benjamini and Hochberg procedure at level δ = 0.05 was performed to check the false discovery rate (FDR) [22] (*m*, the number of tests was equal to 68 after removing the 2 housekeeping genes from the list of 70 miRNAs). However, this procedure does not take into account potential correlation among miRNAs expression levels and thus among *p*-values, the final models were developed as reported below. 

The final multivariable models for ADC and SCC were obtained using the elastic net regularized Cox regression model, setting the mixing parameter to 0.5 and a maximum number of 5 variables to include in each model. This last choice was due to the relatively low number of patients and events in each histotype. We chose the elastic net because of its ability to select miRNAs in the presence of correlated variables. The mixing parameter of 0.5 represents a compromise between LASSO and ridge regression, thus allowing for variable selection as well as parameter shrinkage towards the more contributive variables. Ten-fold cross-validation was used to find the optimal regularization tuning parameter, lambda. We did not impose shrinkage for patient or clinical covariates, e.g., disease stage. The final model retained all independent variables with beta coefficients not equal to zero. Results are only reported in terms of beta regression coefficients as the computation of coefficient standard errors as well as other quantities (e.g., confidence intervals) within the context of Cox elastic net models is still a problematic issue [23]. Performances of the final models were assessed through cross-validation following the procedure proposed by Simon et al. [24]. The two aspects evaluated in the model validation were (1) the ability of the model to separate patients into groups characterized by different risks for the events of interest (i.e., low and high risks) and (2) the discriminant capacity of the model. Thus, leave-one-out cross-validated K–M curves (from now on called cross-validated K–M curves) and leave-one-out cross-validated time-dependent receiver-operating characteristic (ROC) curves (from now on called cross-validated ROC curves) were computed, respectively, to evaluate these two aspects. The permutation test on the cross-validated log-rank statistics was used to compare survival curves; 1000 permutations of the correspondence between different miRNAs and outcome variables and disease stage were performed. In a similar way, a permutation test was used to evaluate whether the area under the ROC curve (AUC) was equal to 0.5.

We performed a permutation test where only the miRNAs were randomly reassigned to evaluate whether a model including standard covariates (e.g., disease stage) and miRNAs had higher prediction accuracy than the standard covariate-only model, as done by Simon et al. [24]. The correspondence between time, event status and covariates in this case was not disrupted. Thus, we tested the null hypothesis that survival and covariates are independent of miRNAs, both according to the log-rank statistics and AUC. All statistical analyses were performed in R version 3.4.1, (R Core Team, Vienna, Austria).

## 3. Results

### 3.1. Patient Characteristics and Outcome

Patients’ characteristics are reported in Table 1. A higher percentage of males and smokers were present in the SCC cohort. The majority of patients had stage I or II disease. In particular, among those with SCC, 50.6% were stage I (of whom 45% stage IA and 55% stage IB), 32.5% were stage II (of whom 30% stage IIA and 70% stage IIB), and 16.9% were stage IIIA. 

Among those with ADC, 70.7% were stage I (54% stage IA and 46% stage IB), 12.1% were stage II (25% stage IIA and 75% stage IIB), and 17.2% stage IIIA. With regard to patient’s treatment, 18% and 12% of SCC patients had chemotherapy and radiotherapy, respectively. Similar percentages were observed for ADC patients, Table 1. Both treatments were significantly associated with stage of disease for both histotypes, results not shown. Overall, 73% (61/83) and 54% (53/99) of SCC and ADC patients, respectively, relapsed or died during follow-up. In particular, 51% (42/83) and 35% (35/99) of SCC and ADC patients, respectively, relapsed or died within 2 years from diagnosis. Median DFS was of 23.2 months (95% CI 15.5–50.0) and 104.0 months (95% CI: 38.4–112.0) for SCC and ADC patients, respectively. Median follow-up was 78.95 months (min–max: 41.8–160.7 months) and 66.3 months (min–max: 41.4–171.1 months) for the SCC and ADC cohorts, respectively. With regard to OS, 63% (52/82) of SCC patients and 41% (41/99) of ADC patients died during follow-up. Median OS was 50 months (95% CI 31.4–76.5) and 112.8 months (95% CI 81.8-NA), respectively. Median follow-up was 90.03 (95% CI 1.5–160.7) and 78.72 (95% CI 41.1–171.1), respectively.

At univariate analysis, a significant association was observed between stage of disease and DFS or OS in both SCC and ADC patients (Appendix A). In particular, a significantly shorter DFS was observed for SCC patients with stage II (HR 1.84, 95% CI: 1.04–3.26, *p* = 0.036) and stage III tumors (HR 3.43, 95% CI: 17.45–86.78 *p* < 0.001) compared to those with stage I disease. With regard to OS, a significantly higher risk of death was observed for patients with stage IIIA disease with respect to earlier stages (HR 6.32, 95% CI: 3.36–11.86, *p* < 0.001). Analogously, in the ADC cohort, patients with stage II or III tumors showed a significantly shorter DFS than those with stage I disease (HR = 2.73, 95% CI: 1.22–6.11, *p* = 0.014 and HR = 6.32, 95% CI: 3.36–11.86, *p* < 0.001, respectively). Similarly, with regard to OS, a significantly higher risk of death was observed for stage IIIA patients with respect to earlier stages (HR 6.32, 95% CI: 2.50–9.93, *p* < 0.001), and a trend was observed for stage II with respect to stage I tumors (HR 2.6, 95% CI: 0.97–6.22, *p* = 0.058) (Table 2). Considering stage as a 5-category variable, stage IIIA compared to IIB patients showed a higher risk of relapse or death that was significant in ADC patients (HR = 3.06, 95%CI: 1.11–8.42, *p* = 0.030) but not in SCC patients (HR = 1.47, 95% CI: 0.70–3.08, *p* = 0.304). Similar results were obtained for OS (HR 3.37, 95% CI 0.96–11.92, *p* = 0.059 and HR 1.40, 95% CI: 0.63–3.10, *p* = 0.404, for ADC and SCC, respectively).

At univariate analyses, chemotherapy and radiotherapy were associated with a worse DFS for SCC patients, Table 2. These results were mainly due to the confounding effect of disease stage. In fact, by adjusting for this covariate, both chemotherapy and radiotherapy resulted associated with a better prognosis (HR = 0.96, 95% CI: 0.46–2.02, *p* = 0.916 and HR = 0.63, 95% CI: 0.27–1.45, *p* = 0.277 for chemotherapy and radiotherapy, respectively). Similarly, by adjusting for disease stage, both treatments were associated with a better OS (HR = 0.93, 95% CI: 0.39–2.22, *p* = 0.868 and HR = 0.56, 95% CI: 0.20–1.54, *p* = 0.260 for chemotherapy and radiotherapy, respectively). With regards to ADC patients, at univariate analysis chemo- and radiotherapy were associated with a poor prognosis, in terms of DFS or OS, Table 2. As before, after adjustment for disease stage both treatments resulted associated with a DFS and OS. In particular, for DFS the adjusted HR was equal to 0.40 (95% CI: 0.18–0.86, *p* = 0.016) for chemotherapy and to 0.45 (95% CI: 0.17–1.18, *p* = 0.106) for radiotherapy.

Similarly, for OS the adjusted HR was equal to 0.24 (95% CI: 0.10–0.61, *p* = 0.002) for chemotherapy and to 0.41 (95% CI: 0.14–1.21, *p* = 0.106) for radiotherapy.

With regard to the other clinical-pathological characteristics of patients, the only significant association was observed in the group of ADC patients in relation to smoking habits: former smokers had a higher risk of relapse or death (HR = 2.76, 95% CI: 1.02–7.46, *p* = 0.045), Table 2.

### 3.2. Selection of Endogenous Reference miRNAs 

Fourteen of the 84 miRNAs available were excluded as they were undetermined in >50% of patients: miR-1-3p, miR-133a-3p, miR-133b, miR-141-3p, miR-200a-3p, miR-203a-3p, miR-208a-3p, miR-215-5p, miR-499a-5p, miR-9-5p, miR-184, miR-206, miR-373-3p, and miR-965, and were the same for ADC and SCC patients. In preliminary analyses, these miRNAs did not show any prognostic potential either with regard to DFS and OS, with and without values imputation. The other miRNAs were evaluated using normFinder to identify invariant miRNAs to use as endogenous reference controls. The most stable miRNAs were miR-221-3p and miR-24-3p for SCC patients and miR-221-3p and miR-126-3p for ADC. These miRNAs were used, together with the exogenous cel-miR-39, in the normalization procedures described in the “miRNA Expression Analysis” section.

### 3.3. Circulating miRNA in Relation to Patient Outcome

Overall, 68 miRNAs were analyzed in relation to DFS (Appendix A) and OS (Appendix A). In univariate analysis, five miRNAs (miR-26a-5p, miR-126-3p, miR-130b-3p, miR-205-5p, and miR-21-5p) were significantly associated with DFS in SCC patients at 5% significance level, whereas two miRNAs (miR-26b-5p and let-7a-5p) showed borderline significance (Table 3A). Of the five miRNAs, none remained significant after adjusting for multiple comparisons at δ = 0.05. Considering a less stringent threshold for δ, i.e., δ = 0.10, miR-26a-5p and miR-126-3p were significant. Of the five miRNAs significantly associated with DFS, four (miR-130b-3p, miR-26a-5p, miR-126-3p, and miR-205-5p) were also significantly associated with OS. Of these, none remained significant after adjusting for multiple comparisons at δ = 0.05 (Table 4A). In ADC patients, only miR-222-3p, miR-22-3p and mir-93-5p were significantly associated with DFS in univariate analysis, whereas miR-19b-3p showed borderline significance. The significance disappeared when we adjusted for multiple comparisons (Table 3B). Moreover, miR-22-3p and miR-19b-3p were significantly associated with OS, whereas miR-195-5p showed borderline significance (Table 4B). The significance disappeared when adjusting for multiple comparisons was done.

Given the high number of biomarkers studied compared to the number of the events of interest, a regularized Cox model including all miRNAs and covariates, e.g., disease stage, was fitted separately for SCC and ADC patients as described in the “Statistical Analysis” section. The final combined model for SCC patients included disease stage and miR-126-3p. With respect to stage I, stages II and III were associated with an increased risk of relapse or death (beta coefficients equal to 0.60 and 1.23, respectively), in particular stage II. miR-126-3p had an independent prognostic role associated with a lower risk of relapse or death (beta coefficient equal to −0.03). 

Performance of the obtained final model was evaluated through cross-validation and following, as outlined in the “Statistical Analysis” section, the procedure by Simon et al. First, we evaluated the ability of the combined model to classify patients according to risk of relapse or death (high versus low based on the median PI values). This aspect was graphically evaluated using leave-one-out cross-validated K–M curves (Figure 1). The same analysis was performed for a model including only stage of disease, the only clinical covariate selected by elastic net among those for age, gender, smoking habit, and type of treatment. Neither of the models proved capable of separating patients into groups characterized by a significantly different risk of relapse or death (*p* = 0.089 and *p* = 0.102 for the combined and stage-only models, respectively). We also evaluated the predictive accuracy of both models. Figure 2 shows the cross-validated time-dependent ROC curves of the combined and stage-only models. Their AUCs were 0.681 and 0.720, respectively, and both were significantly different from 0.5 (*p* < 0.001 for both). Finally, formally testing for the added predictive accuracy induced by the combined model over the stage only one did not reveal a statistically significant improvement in terms of either model’s ability to separate patients into risk groups (*p* = 0.928) and AUC (*p* = 0.816). With regard to ADC histotype, the regularized Cox model did not identify any miRNAs. Disease stage was the only variable significantly associated with DFS. No other patient’s information (e.g., age, gender, and smoking habit) was significantly associated to DFS in the multivariable model. With regard to the secondary endpoint (OS), no miRNAs were identified as significant by the regularized Cox model after adjusting for stage of disease. This was true for both histotypes. Disease stage was the only independent factor associated with OS.

## 4. Discussion

Early-stage NSCLC represents the minority of all NSCLC diagnoses and can be curatively treated by surgery alone or by adjuvant chemotherapy. In particular, patients with stage IA NSCLC generally undergo surgery alone, but a subgroup of these could benefit from adjuvant chemotherapy. Conversely, patients with stage IB-IIIA usually receive both surgery and adjuvant chemotherapy and/or radiotherapy, but some may receive unnecessary treatments with potential side-effects [3,25,26]. As approximately 30–50% of patients with early-stage NSCLC relapse within five years of initial treatment, the search to find prognostic biomarkers capable of identifying those at higher risk of relapse is warranted. 

In this study, we analyzed a panel of circulating miRNAs separately in two cohorts of patients with early-stage NSCLC, ADC or SCC. With regard to patients and clinical covariates, disease stage was confirmed to be the strongest predictor of DFS in both patient groups. Moreover, after adjusting for stage of disease, the only covariate resulting significant among age, gender, smoking habit, and type of treatment, miR-126-3p showed an independent prognostic role in the SCC cohort. However, although this miRNA seemed promising, its inclusion in a model containing disease stage did not substantially improve prediction accuracy with respect to a model including only disease stage. In the ADC group no miRNAs remained significantly associated with prognosis after adjusting for disease stage. 

In our study stage I ADC patients showed a better prognosis with respect to stage I SCC patients. Although the number of cases was small, this is in agreement with a previous report, in which the authors demonstrated a better prognosis of ADC with respect to SCC patients, for stage I disease, whereas the prognosis of ADC was worse considering the advanced stages [27].

Numerous authors have addressed the issue of prognostic biomarkers in early-stage NSCLC by studying circulating miRNAs. However, results are difficult to compare for a number of reasons and no definitive conclusions can be drawn. Heterogeneity among studies ranges from the population of interest (e.g., SCC versus ADC analyzed together or separately), outcome measure (e.g., DFS or OS), biological sample available (e.g., serum versus plasma), normalization techniques used for miRNA expression levels, statistical technique used to analyze the data and presence of a validation analysis, presence of detailed clinical and histological information, and study sample size.

Some studies analyzed specific preselected miRNAs in relation to patient outcome [28,29,30]. miR-98 [29], miR-195 [28], and miR-34 [30] were significantly associated with prognosis, DFS, or OS, depending on the study. Other reports focused on a large panel of miRNAs [20,21,31,32], using different miRNA panels and platforms. Results from these studies revealed different miRNA signatures associated with patient outcome, DFS, OS or time to recurrence. In particular, a signature of four miRNAs (miR-486, miR-30d, miR-1, and miR-499) was consistently associated with OS in both a training and validation set of early-stage NSCLC [20]. In another study, two distinct signatures were found to predict outcome in early-stage ADC (miR-155-5p, miR-223-3p, and miR-126-3p) and SCC (miR-20a-5p, miR-152-3p, and miR-199a-5p) patients separately [31]. In other experiences, starting from a miRNA profile analysis, miR-150 [21], and miR-324-3p [32] were significantly associated with prognosis in early-stage NSCLC. Several differences emerge from the studies in the literature that may explain the heterogeneity of results. For example, the majority analyzed the prognostic value of circulating miRNAs in NSCLC patients without distinguishing between ADC and SCC histotypes [20,21,28,29,30]. Moreover, although all of the studies considered early-stage NSCLC, some focused on stage I [32], some on stages I and II [21], and some on stage I-IIIA [20,28,30,31]. Another difference in the studies is the type of reference miRNAs used for the normalization process. As there are still no universally validated housekeeping miRNAs, some studies used the synthetic exogenous cel-miR-39 [28,32] or short U6 [31] as normalizer, and some used a normalization process based on healthy donor samples [20]. 

Our study has a number of strengths. First, given that ADC and SCC represent two distinct tumors with different biology, we considered the ADC and SCC cohorts separately. We also selected the reference miRNAs separately in the two histotypes. We performed a selection of reference miRNAs applying the (default) NormFinder algorithm, using both endogenous and exogenous references to normalize data. The analyses were carried out on two separate early-stage NSCLC cohorts with a homogeneous follow-up and end point definition. Finally, as external independent cohorts of SCC and ADC patients were not available to validate our prognostic models, we performed an internal evaluation using a cross-validation technique. Using Simon et al.’s method [24] we also compared the predictive value of miRNAs plus standard prognostic factors with that of standard prognostic factors alone, e.g., disease stage. Adjusting for disease stage, we found that miR-126-3p was independently associated with DFS in SCC patients, whereas no significant association was observed in the ADC cohort. However, a model considering both miR-126-3p and disease stage did not show notably better predictive accuracy than the model containing disease stage alone. The prognostic role of miR-126 was evaluated in two recent meta-analyses [33,34]. Dong et al. [33], looking at different types of cancers, found that high levels of miR-126 played a favorable role in OS that was particularly significant in tumors of the digestive system (HR 0.70, 95% CI: 0.59–0.83) and the respiratory tract (HR 0.71, 95% CI: 0.59–0.85). Conversely, the meta-analysis by Zheng et al. [34] focused on NSCLC alone. In their study, four studies were eligible for pooled analysis, all agreeing with our results that low miR-126 tissue levels were associated with poor prognosis [35,36,37]. Interestingly, in a study by Donnem et al. [38], tissue miR-126 expression was found to be a significant negative prognostic factor in both univariate and multivariate analyses. In particular, after stratification by histology, miR-126 only remained significant in SCC patients (HR 3.1, 95% CI: 1.7–5.6, *p* < 0.001), in agreement with our findings. With regard to circulating miR-126, plasma levels of the miRNA were found to be decreased in NSCLC patients [39] and associated with a higher risk of relapse in ADC patients [31]. Moreover, low serum miR-126 levels were associated with worse prognosis in patients with mesothelioma [40].

Numerous reports have demonstrated that miR-126 may function as an important regulatory factor in the development of NSCLC. In particular, its upregulation may reduce the expression of the target gene PIK3R2, influencing the PTEN/PIK3CA/AKT pathway and reducing the proliferation, migration and invasion of cells [41]. Moreover, its decrease could enhance the adhesion, migration and invasion properties of NSCLC cells by increasing Crk protein expression [42]. miR-126 may also inhibit NSCLC cell proliferation by targeting EGFL7 [43].

Our study also has some limitations. First, the panel of miRNAs we analyzed was not specifically selected for NSCLC but rather was a commercial panel of miRNAs usually identifiable in plasma/serum. We also did not have two independent SCC and ADC cohorts that could serve as external validation sets. However, we still chose to perform internal validation as a way to corroborate our findings. Despite internal validation is not able to reflect the many sources of variability present in clinical, it can still represent a valid tool to evaluate performance of the model at hand. Indeed, a model showing good performances in cross-validation is more likely to perform well on future patient samples than one that does not. In our study, some “promise” is evident at least for the squamous cell histotype and an external validation as well as a more comprehensive analysis of other potentially prognostic biomarkers will be performed in an ongoing multicenter prospective study. 

Our study demonstrated that clinical stage remains the strongest prognostic indicator in early-stage NSCLC, and that circulating miR-126-3p plays its independent role when considered with disease stage in SCC patients. However, the model including both factors did not show a higher predictive accuracy compared to the model with disease stage alone. Further research is warranted in larger and independent patient cohorts to investigate the role of miR-126-3p, other miRNAs and standard covariates, and to search for other molecular factors to improve the prediction capability of prognostic models.

## Figures and Tables

**Figure 1 jcm-08-00131-f001:**
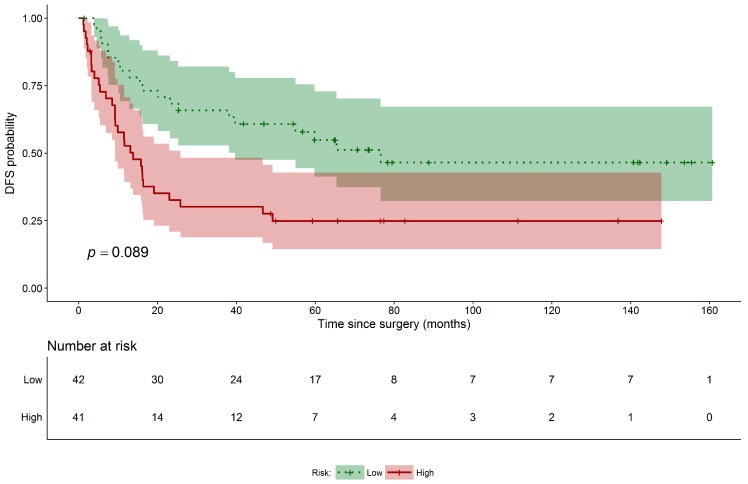
DFS Kaplan–Meier curves for SCC patients. Disease-free survival risk table and leave-one-out cross-validated Kaplan–Meier curves for SCC high (red solid line) and low risk (green dotted line) patients groups along with their 95% confidence interval (accordingly shaded areas); permutated log-rank *p*-value for the hypothesis testing of equality of curves between groups is also reported.

**Figure 2 jcm-08-00131-f002:**
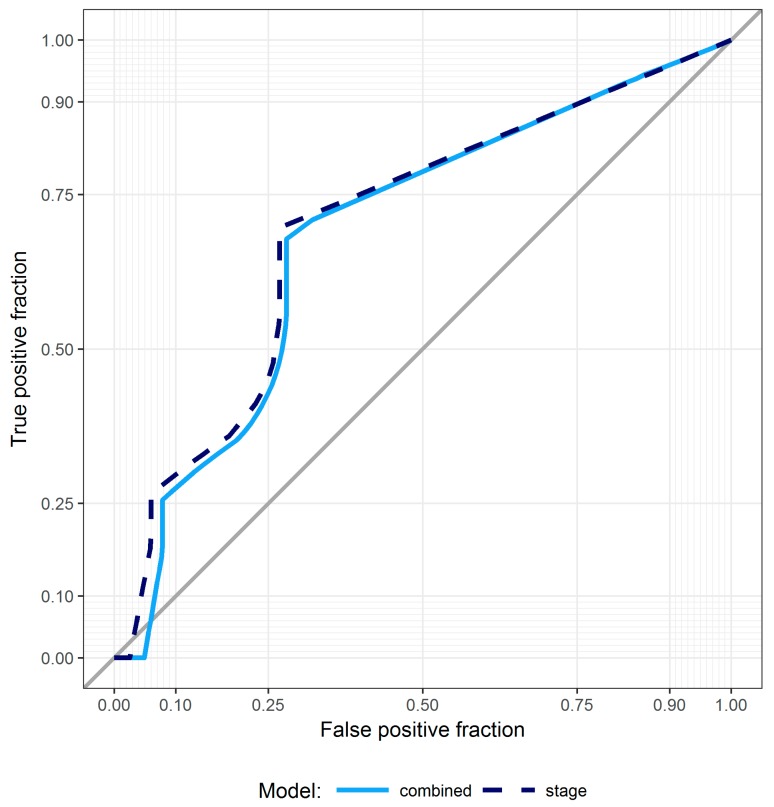
ROC curves for SCC patients. Cross-validated time-dependent ROC curves for the combined (cyan solid line) and stage-only (dashed blue line) models in SCC patients; bisector gray line identifies a purely random discrimination.

**Table 1 jcm-08-00131-t001:** Patient characteristics of the analyzed cohorts.

	SCC (*n* = 83)			ADC (*n* = 99)		
	*n*	(%)	Relapse or Death	*n*	(%)	Relapse or Death
**Gender**						
Female	11	(13.25)	8	40	(40.40)	16
Male	72	(86.75)	53	59	(59.60)	37
**Mean age at surgery ± SD (years)**	68.47 ± 7.53			67.58 ± 9.09		
**Smoking habit ^a^**						
Non-smoker	-	-		17	(18.68)	5
Ex-smoker	22	(27.85)	14	29	(31.87)	18
Current smoker	57	(72.15)	44	45	(49.45)	26
**Stage of disease**						
I	42	(50.60)	26	70	(70.71)	28
II	27	(32.53)	22	12	(12.12)	8
IIIA	14	(16.87)	13	17	(17.17)	17
**Adjuvant chemotherapy ^a^**						
No	67	(81.71)	50	84	(84.85)	41
Yes	15	(18.29)	11	15	(15.15)	12
**Adjuvant radiotherapy**						
No	73	(87.95)	53	89	(89.90)	44
Yes	10	(12.05)	8	10	(10.10)	9

SCC: squamous cell carcinoma; ADC: adenocarcinoma; SD: standard deviation. ^a^ Number may not add up to the total number of subjects due to missing data.

**Table 2 jcm-08-00131-t002:** Risk of relapse or death in squamous cell carcinoma (SCC) and adenocarcinoma (ADC) patients in relation to clinical-pathological characteristics.

	DFS				OS			
	SCC		ADC		SCC		ADC	
	HR (95% CI)	*p*	HR (95% CI)	*p*	HR (95% CI)	*p*	HR (95% CI)	*p*
**Gender**								
Female	1		1		1		1	
Male	1.13 (0.53–2.38)	0.752	1.56 (0.86–2.83)	0.139	0.90 (0.41–2.01)	0.802	1.95 (0.95–4.02)	0.069
**Age at surgery, years ^a^**	1.01 (0.97–1.04)	0.746	1.01 (0.98–1.04)	0.494	1.03 (0.99–1.06)	0.154	1.03 (0.98–1.07)	0.066
**Smoking habit**								
Non-smoker	-		1		-		1	
Ex-smoker	1		2.76 (1.02–7.46)	0.045	1		2.07 (0.67–6.44)	0.209
Current smoker	1.46 (0.80–2.70)	0.220	1.88 (0.72–4.97)	0.200	1.48 (0.75–2.93)	0.259	1.65 (0.56–4.91)	0.366
**Stage of disease**								
I	1		1		1		1	
II	1.84 (1.04–3.26)	0.036	2.73 (1.22–6.11)	0.014	1.70 (0.92–3.15)	0.091	2.6 (0.97–6.22)	0.058
IIIA	3.43 (1.74–6.78)	<0.001	6.32 (3.36–11.86)	<0.001	2.73 (1.32–5.69)	0.007	6.32 (2.50–9.93)	<0.001
**Adjuvant chemotherapy**								
No	1		1		1		1	
Yes	2.30 (1.20–4.39)	0.012	1.88 (0.87–4.09)	0.110	1.03 (0.53–1.98)	0.936	0.72 (0.34–1.53)	0.391
**Adjuvant radiotherapy**								
No	1		1		1		1	
Yes	1.85 (0.90–3.80)	0.096	1.27 (0.50–3.23)	0.621	1.42 (0.67–2.99)	0.358	1.16 (0.49–2.72)	0.733

^a^ One-year increment. DFS: disease-free survival; OS: overall survival; SCC: squamous cell carcinoma; ADC: adenocarcinoma.

**Table 3 jcm-08-00131-t003:** miRNAs significantly associated with disease-free survival (DFS) at univariable analysis.

**A. SCC**				
**Name**	**HR**	**95% CI**	***p*-Value**	**(j/m) × δ**
mir-26a-5p	0.57	0.4–0.80	0.00102	0.00074
mir-126-3p	0.57	0.4–0.80	0.00147	0.00147
mir-130b-3p	0.74	0.59–0.92	0.00775	0.00221
mir-205-5p	1.15	1.02–1.29	0.02479	0.00294
mir-21-5p	0.66	0.44–0.97	0.03552	0.00368
mir-26b-5p	0.78	0.61–1.0	0.05318	0.00441
let7a-5p	0.76	0.58–1.01	0.05588	0.00515
**B. ADC**				
**Name**	**HR**	**95% CI**	***p*-Value**	**(j/m) × δ**
mir-222-3p	1.37	1.06–1.76	0.01628	0.00074
mir-22-3p	1.23	1.03–1.49	0.02641	0.00147
mir-93-5p	1.41	1.02–1.99	0.03713	0.00221
mir-19b-3p	1.35	0.99–1.62	0.05898	0.00294

DFS: disease-free survival; SCC: squamous cell carcinoma; HR: hazard ratio; ADC: adenocarcinoma; m: number of tests; δ: FDR level.

**Table 4 jcm-08-00131-t004:** miRNAs significantly associated with overall survival (OS) at univariable analysis.

**A. SCC**				
**Name**	**HR**	**95% CI**	***p*-value**	**(j/m) × δ**
mir-130b-3p	0.74	0.59–0.92	0.00753	0.00074
mir-26a-5p	0.62	0.43–0.88	0.00767	0.00147
mir-126-3p	0.62	0.43–0.9	0.01242	0.00221
mir-205-5p	1.16	1.01–1.33	0.04138	0.00294
**B. ADC**				
**Name**	**HR**	**95% CI**	***p*-value**	**(j/m) × δ**
mir-22-3p	1.29	1.05–1.59	0.01726	0.00074
mir-19b-3p	1.33	1.01–1.76	0.04331	0.00147
mir-195-5p	1.32	0.99–1.77	0.05904	0.00221

OS: overall survival: SCC: squamous cell carcinoma; HR: hazard ratio; ADC: adenocarcinoma; m: number of tests; δ: FDR level.

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
