# Peer review of "Prognostic Role of Circulating miRNAs in Early-Stage Non-Small Cell Lung Cancer"

_jcm, 2019, doi:10.3390/jcm8020131_

Reviewer 1 Report

The evaluation of miRNAs in LC patients is still an attractive option for future menagement of this most prevalent human malignancy, The strongest point of the followed study is a large set of serum specimens collected from eraly stage LC.

Due to reviewer's responsibility, I would like to point some comments:

In my opinion the major limitation of this study is a low number of analyzed miRNAs (84 molecules), therefore a lot potenital prognostic markers are missed. I realize that re-analysis will change the all results definitely, however authors should explain why they decided to use this miRNA panel. I believe that further analysis with this precious metarial would bring a outstanding results/

It will be interesting by adding additional short paragraph to results section on analysis only patients with stage I-II and excluding patients treated with either post-operative CTH or RTH.

The introduction is written by manner similar to discussion. I suggest to not quote particular authors, but rather first, focus on the role of miRNAs in LC and their prognostic significance, second why these molecules are considered as putitative prognostic markers of LC. 

Some information about patients can be added, e.g. newly diagnsed/recurrence, ECOG stage, pack-years, and T and N stage. Maybe some of these affect to either DFS or OS.

Author Response

Reviewer 1

The evaluation of miRNAs in LC patients is still an attractive option for future menagement of this most prevalent human malignancy, The strongest point of the followed study is a large set of serum specimens collected from eraly stage LC.

Due to reviewer's responsibility, I would like to point some comments:

1-In my opinion the major limitation of this study is a low number of analyzed miRNAs (84 molecules), therefore a lot potenital prognostic markers are missed. I realize that re-analysis will change the all results definitely, however authors should explain why they decided to use this miRNA panel. I believe that further analysis with this precious metarial would bring a outstanding results/

 Reply: we agree with the reviewer with regard to this point. However, this was an exploratory study in which we chose to analyse a restricted panel of miRNAs on the basis of literature results. For this purpose, we chose to use the “Human Serum & Plasma miScript miRNA PCR Array”, designed to detect miRNAs expressed in serum, plasma, and other extracellular body fluids, reported to be differentially regulated in serum or plasma samples in cancer. They comprise most of the miRNAs for which there were evidences of their role in lung cancer. However, we designed a prospective multicenter cohort study (funded by TRANSCAN Joint Transnational Call 2016 (JTC 2016)), actually ongoing, to validate our results and to potentially expand them by evaluating other miRNAs and biomarkers.This point has been discussed in the text (page 13, lines 483-493).

 2-It will be interesting by adding additional short paragraph to results section on analysis only patients with stage I-II and excluding patients treated with either post-operative CTH or RTH.

 Reply:  We thank the reviewer for this observation and indeed subgroup analysis would be very interesting. However, the main reason why we did not report these analyses follows the same logic used for analysing the entire cohort. That is, to avoid as much as possible the possibility of incurring in false positives results and in the identification of unstable models due to small sample sizes and a concurrent small number of the events of interest. For subgroup analyses as for those on the entire cohort of patients, the main purpose is still to find new prognostic biomarkers. For doing so all the computational procedures reported in the statistical analysis section, including cross-validation, have to be performed from scratch for each subgroup. Another problem that can arise when the sample size gets smaller and smaller is a worsening in performances of the cross-validation techniques for evaluating model’s prediction accuracy. These two reasons desisted us from performing subgroups analyses.

 3-The introduction is written by manner similar to discussion. I suggest to not quote particular authors, but rather first, focus on the role of miRNAs in LC and their prognostic significance, second why these molecules are considered as putitative prognostic markers of LC. 

 Reply: We thank the referee for this suggestion. We have modified the introduction following the referee’s indications

 4-Some information about patients can be added, e.g. newly diagnsed/recurrence, ECOG stage, pack-years, and T and N stage. Maybe some of these affect to either DFS or OS.

 Reply: With regard to this point, all patients had a new diagnosis of early-stage NSCLC and all had an ECOG PS of 0 or 1 making the patients’ cohort quite homogeneous to this respect (all candidates to a surgical treatment). With regard to T and N, these parameters are already taken into account in stage classification. In the analyses, we reported and used disease stage instead of its specific components (i.e. T, N, M) due to the correlation often encountered among them, and to avoid to analyse too small subgroups. Finally, this is a retrospective cohort study for which clinical and demographic information were retrieved from medical charts. As a consequence, information on smoking habit was present in terms of never, former or current smoker, and not with regard to pack-years.

Reviewer 2 Report

The authors investigated circulating miRNAs as potential prognostic markers in NSCLC patients. They used a panel containing 84 miRNAs primers relevant in biofluids, but not optimized for lung cancer. None of the miRNAs were significantly associated with prognosis after adjusting for multiple comparisons. Stage is a well-known prognostic factor and was included in the final multivariable model. The predictive accuracy for stage combined with miR-126-3p revealed AUC of 0.68, whereas the stage alone showed AUC of 0.72. By using leave-one-out cross validation, the model could not separate the patients into groups based on risk of relapse or death. Although miR-126-3p may be an interesting miRNA and may have a prognostic role in SCC, the results are not convincing and should be tested in an external validation data set before publishing. 

Minor:

2.2 miRNA expression analysis

Normalization: the term reference genes should be used instead of housekeeping genes as no housekeeping gene exist in serum.

More information about the method could be provided. Are the miRNAs run in duplicates or triplicates? Was the RNA quantity measured, and what was the RNA input in the analysis?

More information regarding the material could be provided. Storage temperature, duration of freezing, number of freeze/thaw cycles and handling of the samples in general are lacking.  

2.3 statistical analyses

Why were miRNAs with >50% of undetermined values excluded?

By imputing the missing values (K-nearest Neighbor method), the missing values are assumed to be technical failure and not biological variation. Biological important miRNAs may be detected in only a subset of the patients.By excluding miRNA detected only in a subset of the samples or by imputing missing values, biologically important miRNAs may have been lost.

Figure 2. The predictive accuracy was shown with ROC curves for the combined model and for stage alone. How did miR-126-3p perform alone? The prognostic role of miR-126-3p is not clear/ convincing since it’s not externally validated and was not significant after correcting for multiple testing.

It is not clear which covariates other than disease stage included in the Cox model.

3.1 Patients characteristics and outcome

The results from this section would be easier to read if the text could be reduced or minimized to the most important findings. Stage is a well-known prognostic factor and some of the comparisons could be moved to supplementary. However, the prognostic differences detected in ADC and SCC are not explained and could be discussed. Only results from part 3.3 are emphasized in the discussion part.

Author Response

Reviewer 2

1-The authors investigated circulating miRNAs as potential prognostic markers in NSCLC patients. They used a panel containing 84 miRNAs primers relevant in biofluids, but not optimized for lung cancer. None of the miRNAs were significantly associated with prognosis after adjusting for multiple comparisons. Stage is a well-known prognostic factor and was included in the final multivariable model. The predictive accuracy for stage combined with miR-126-3p revealed AUC of 0.68, whereas the stage alone showed AUC of 0.72. By using leave-one-out cross validation, the model could not separate the patients into groups based on risk of relapse or death. Although miR-126-3p may be an interesting miRNA and may have a prognostic role in SCC, the results are not convincing and should be tested in an external validation data set before publishing. 

 Reply: We understand and agree with the reviewer comment with regard to the need of validating our results in an independent cohort of patients. However, due to a relatively small sample size, we chose to perform internal validation  to corroborate our findings. Indeed, even if internal validation can not reflect many of the sources of variability present in clinical practice outside the context where the present research is conducted, it can still represent a valid tool to evaluate performance of the model at hand. Indeed, a model that performs well in cross-validation is more likely to perform well on future patient samples than one that does not. In our study, some “promise” was obtained at least for the squamous cell histotype. An external validation as well as a more comprehensive analysis of other potentially prognostic biomarkers will be performed in a multicenter prospective study (still ongoing) funded by TRANSCAN Joint Transnational Call 2016 (JTC 2016). These aspects are now explained in the discussion section (page 13, lines 483-493)

 Minor:

2- 2.2 miRNA expression analysis

Normalization: the term reference genes should be used instead of housekeeping genes as no housekeeping gene exist in serum.

 Reply: The term has been substituted

 3- More information about the method could be provided. Are the miRNAs run in duplicates or triplicates? Was the RNA quantity measured, and what was the RNA input in the analysis?

 Reply: Further information have been added (page 3, lines 108-112)

 4- More information regarding the material could be provided. Storage temperature, duration of freezing, number of freeze/thaw cycles and handling of the samples in general are lacking.  

 Reply: Further information have been added (page 3, lines 104-105)

 5- 2.3 statistical analyses

Why were miRNAs with >50% of undetermined values excluded?

By imputing the missing values (K-nearest Neighbor method), the missing values are assumed to be technical failure and not biological variation. Biological important miRNAs may be detected in only a subset of the patients.By excluding miRNA detected only in a subset of the samples or by imputing missing values, biologically important miRNAs may have been lost.

 Reply: We thank the reviewer and agree with this observation. Before definitely excluding miRNAs with more than 50% of undetermined values from final analyses, some preliminary analyses were performed. In particular, the Chi-square test and the Fisher Exact test, as appropriate, were used to test the hypothesis of independence between presence of undetermined values and DFS (presence or absence of relapse or death) or OS (presence or absence of death). Moreover, to take into account the time-dependence of the two endpoints, DFS and OS, a Cox univariate analysis was performed with and without imputing values for the miRNAs with more than 50% of undetermined values. No statistically significant associations were found. Moreover, the percentage of undetermined values was much further 60% in both the group of patients with relapse/death and in that without relapse/death. We decided then to exclude these biomarkers in light of their low prognostic potential and to reduce data dimensionality and eventually noise a little bit. This point was discussed at page 4, lines 132-134 and at page 9 and lines 285-287.

 6- Figure 2. The predictive accuracy was shown with ROC curves for the combined model and for stage alone. How did miR-126-3p perform alone? The prognostic role of miR-126-3p is not clear/ convincing since it’s not externally validated and was not significant after correcting for multiple testing.

 Reply: We share the reviewer interest on the role of miR-126-3p. However, the statistical approach we followed (see paper by Simon et al Brief Bioinform. 2011, 12, 203-214) works a bit differently. Here is a brief explanation. The entire procedure (i.e. elastic-net plus cross-validation) was run to evaluate and define a model for: i) only the miRNAs, ii) miRNAs plus standard covariates, iii) only standard covariates. Thus, three final models were obtained as well their performance measures. This approach was followed for each cancer histological type and endpoint. The model evaluating only the role of miRNAs for example for the squamous cell carcinoma, contained miR-126-3p as well as miR-130b-3p, miR-26a-5p, and miR-146a-5p and showed performance worse (i.e. AUCs below 0.5) than the covariates only model or the miRNAs plus covariates model. However, the miRNAs only model results from an unadjusted (for clinical covariates such as, stage) analysis. For this reason and in light of the fact that one of the important clinical needs is to find new biomarkers that show a consistently higher predictive power as compare to standard covariates, we decided to not report these findings

Note: in the case of the squamous histotype, the covariates only model contained only disease stage as it was the only variable selected by the algorithm among other such as, type of treatment, smoking habit, gender, and age. The miRNAs and covariates model contained stage and miR-126-3p, the only one selected from the model.  

On the other hand, as already stated, we agree that the best validation would be that performed on a completely independent external cohort where data are collected from Institutions and laboratories other than those where the developmental study is conducted.

However, for an exploratory study such as this, an internal validation can give some important insight. The development of effective prognostic signatures is commonly a multi-stage process, starting with developmental studies and in which methods for internal validation play generally an initial important role.

Splitting a sample in training and validation set (i.e. validation through sample splitting technique where the samples derives from the same source where same researchers handle the biospecimens and performed molecular analyses) as it is performed in many studies in the literature, has been shown to produce potentially misleading indication of predictive accuracy, especially when sample size is limited (see as an example the paper from Molinaro et al. Bioinformatics 2005;21:3301-07). For these reasons in our study we preferred to retain all information available in the entire cohorts of SCC and ADC patients and to perform cross-validation using a resampling technique that generally present a better bias-variance trade-off in term of ability to appropriately evaluate model performance. Even if miR-126-3p certainly deserve to be further validated in an external set of patients perhaps in combinations with other miRNAs or biomarkers with prognostic potential, it emerged as an independent prognostic factor in this study using a rigorous methodology. Multiple testing correction has been reported as an additional piece of information to say that caution has to be paid in the interpretation of the results; however, this has not to be weighted to much as the correction assumes independence of the p-values (text added at page 4, lines 146-148). For this reason our final model was obtained through elastic-net methodology that better performs when correlation is present among the studied biomarkers.

 7- It is not clear which covariates other than disease stage included in the Cox model.

 Reply: The final model for SCC patients included only disease stage and miR-126-3p. These variables were the only two selected from the elastic-net procedure reported in the statistical analysis section.

 8- 3.1 Patients characteristics and outcome

The results from this section would be easier to read if the text could be reduced or minimized to the most important findings. Stage is a well-known prognostic factor and some of the comparisons could be moved to supplementary. However, the prognostic differences detected in ADC and SCC are not explained and could be discussed. Only results from part 3.3 are emphasized in the discussion part.

 Reply: We thank the referee for the suggestions. The figure relative to stage as prognostic factor is already a supplementary file. With regard to the difference between ADC and SCC, we observed, only for stage I, a better outcome of ADC with respect to SCC. This is in agreement with previous report in which a better outcome of ADC patients was observed considering stage I tumors, whereas ADC showed a worse prognosis in patients with advanced disease (Meng F et al, J Cell Physiol 2018). We have added a comment in the discussion (page 12, lines 420-424)

Round  2

Reviewer 2 Report

This study has several strengths such as sample size, method and normalization. These parameters are often not sufficiently covered in many published studies. Unfortunately, the majority of studies on circulating miRNAs as biomarkers suffer from a widespread lack of reproducibility. The most important finding in this study is not very convincing or strong. A microRNA profiling revealed that one microRNA (miR-126-3p) may have prognostic potential in SCC when combined with stage. The AUC was 0.68 for this model, which is not very high. In an external cohort, the model will usually not perform as well as in the original cohort. Taken into account that stage alone revealed AUC of 0.72, it is not obvious to me that miR-126-3p combined with stage will be an attractive model for prediction of prognosis. 

Author Response

We thank the referee for this comment. We think that it is always difficult to predict how one or more biomarkers will perform on different independent datasets. This, for a number of reasons (e.g. changes in patients mix in the new cohort, different operators and laboratories, size of the independent cohort and availability of information on all the potential confounding factors, etc) and also for biomarkers that appear very promising at the beginning.

We are aware that, in general, one biomarker might not have enough predictive potential compared with standard and strong risk factors such as, disease stage. At the same time it might happen that when combined with other biomarkers, also of different nature, that specific biomarker can offer a great contribution. The identified miRNA, miR-126-3p, seems to have a role in this setting, and some data of literature are in line with this hypothesis (Donnen T et al, Cancer 2011; Markou A et al, Lung Cancer 2013; Tomasetti M et al, Clin Biochem 2012). In this study we analysed only on a restricted number of miRNAs. This might have limited the possibility to find a signature with a greater predictive accuracy. A deeper investigation as well as validation of the role of miRNA-126-3p (by a mirnome analysis) and other types of biomarkers will be performed in the mentioned prospective study that we have ongoing.
